# Muharaf: Manuscripts of Handwritten Arabic Dataset for Cursive Text Recognition

**Mehreen Saeed**[1]  **Adrian Chan**[1]  **Anupam Mijar**[1]  **Joseph Moukarzel**[2]
**Georges Habchi**[2]  **Carlos Younes**[2]  **Amin Elias**[3]  **Chau-Wai Wong**[1]  **Akram Khater**[1]

[1]North Carolina State University,    [2]Holy Spirit University of Kaslik
[3]Lebanese Association for History

{mehreen.mehreen,adrian27513,aamijar230}@gmail.com
{josephmoukarzel,georgeshabchi,carlosyounes}@usek.edu.lb
a.elias@lahlebanon.org
{chauwai.wong,akhater}@ncsu.edu

## Abstract

We present the Manuscripts of Handwritten Arabic (Muharaf) dataset, which is a machine learning dataset consisting of more than 1,600 historic handwritten page images transcribed by experts in archival Arabic. Each document image is accompanied by spatial polygonal coordinates of its text lines as well as basic page elements. This dataset was compiled to advance the state of the art in handwritten text recognition (HTR), not only for Arabic manuscripts but also for cursive text in general. The Muharaf dataset includes diverse handwriting styles and a wide range of document types, including personal letters, diaries, notes, poems, church records, and legal correspondences. In this paper, we describe the data acquisition pipeline, notable dataset features, and statistics. We also provide a preliminary baseline result achieved by training convolutional neural networks using this data.

## 1  Introduction

Modern standard Arabic has more than 400 million native speakers worldwide and is the official language of 24 sovereign countries as of 2024 [36]. Arabic is not only widely spoken but also has a vast collection of historical manuscripts spanning rich literary traditions, poetry, philosophy, and scientific writings. The British Library alone has a massive collection of almost 15,000 works in 14,000 volumes of Arabic manuscripts [33]. A highly accurate optical character recognition (OCR) system for handwritten historic Arabic manuscripts will make these documents accessible to a global community of researchers, historians, literary scholars, linguists, and genealogists.

In the past decade, handwritten text recognition (HTR) has made significant progress through the use of deep neural networks [35, 34, 5, 17, 22]. Unlike traditional HTR systems that employ handcrafted features, these networks are data-hungry and require significant amounts of training data to learn, generalize, and be deployed in real-world scenarios. For Arabic HTR, there are unique challenges involved. The Arabic script is cursive and involves varying letter shapes depending on their positions within a word. Moreover, the harakat and diacritics of the Arabic script add to the difficulty of the task. The scarcity of public datasets, compounded by their relatively small sizes, further exacerbates the challenges.

We created the Manuscripts of Handwritten Arabic (Muharaf[1]) dataset of fully annotated and transcribed 1,644 images to train and evaluate an HTR system for Arabic handwritten historical manuscripts. The document images for this dataset were acquired from the archives of the Phoenix

---

[1]"Muharaf" is Arabic for "typeface".

38th Conference on Neural Information Processing Systems (NeurIPS 2024) Track on Datasets and Benchmarks.

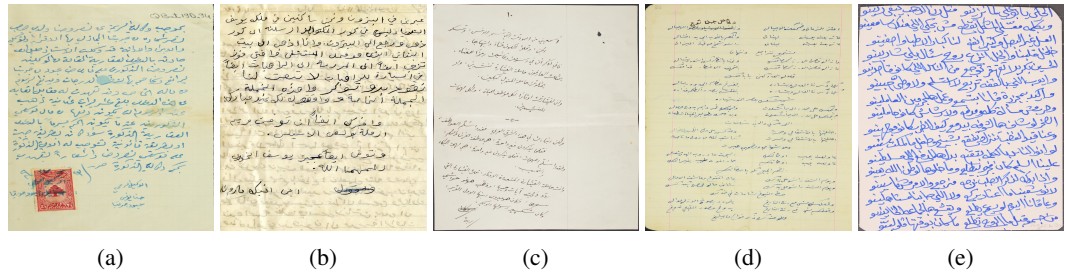

Figure 1: Sample images of the Muharaf dataset from (a) USEK Al Batroun Collection, (b) KCLDS El-Khouri Letters Collection, (c) KCLDS Ameen Rihani Collection, (d) USEK Hanna Moussa Collection, and (e) USEK Joseph El Hachem Collection.

Center for Lebanese Studies at Holy Spirit University of Kaslik (USEK) and the Khayrallah Center for Lebanese Diaspora Studies (KCLDS) at North Carolina State University (NC State). In the preliminary phase of the data collection pipeline, experts in archival Arabic annotated and transcribed the individual text lines in the document images. In the main phase, we leveraged deep learning to predict the texts of the document pages, which was then manually corrected by experts.

The Muharaf dataset can be used not only in HTR systems but also in other text-related tasks such as text-line segmentation, layout analysis, writer identification, style classification, and more. While Muharaf is not customized for training a page layout detection system, it contains the annotation and labeling of some basic page elements. For example, graphics, page numbers, floating regions, crossed-out text, paragraph separators, and signature areas have all been marked and annotated.

The Muharaf dataset consists of a diverse set of images, ranging from individual personal letters, poems, and dialogues to legal consensus records, correspondences, and church records. The manuscripts date from the early 19th century to the early 21st century. There are a total of 1,644 images, 36,311 text lines, and 4,867 text regions including main text regions, headings, and floating text regions. The quality of page images varies, from writing on a clean white background to illegible sentences on creased pages with ink bleeds. A major part of the dataset with 1,216 images, will be made publicly available and the remaining 428 images will be distributed under a proprietary license with permission from the owner. A few samples from the public portion of this dataset are shown in Figure 1. We used OpenAI's GPT APIs [25] to generate English summaries and keywords corresponding to each manuscript page, which we provide for the general interest of the research community.

In this paper, we also present the results of training a convolutional neural network (CNN) based system of start, follow, and read (SFR) networks [35]. Each network can easily fit in an 8 GB graphical processing unit (GPU) card, making it ideal for deployment in a low-resource setting. Using a similar setup, an HTR system for languages based on the Arabic script like Urdu, Farsi, and Pashto can also be developed. Such systems can be initially trained using the Muharaf dataset and subsequently adapted and fine-tuned for the respective languages.

The rest of this paper is organized as follows. Section 2 outlines the relevant characteristics of the Arabic script and Section 3 overviews the existing Arabic datasets for OCR/HTR. Section 4 describes the pipeline for collecting data and Section 5 details the dataset features, formats, and statistics. Section 6 provides the baseline results from our preliminary experiments on HTR. Section 7 concludes the paper and discusses the limitations of this work along with future directions.

## 2    Characteristics of Arabic script in Muharaf

The Arabic alphabet is believed to have its roots in the Nabatean alphabet from the third century [12]. The study of the origins of Arabic script and its evolution from classical Arabic to modern standard Arabic is a huge undertaking and still a subject of open scientific study and debate [33]. For example, the early Arabic writings did not contain dots (ijam) but later the script evolved to include them. The Muharaf dataset has writing samples from the early 19th century to the early 21st century, a period during which the use of dots was well established. Most of Muharaf's samples are in the Arabic script Ruq'ah. Ruq'ah became the most common writing style in the late Ottoman period and early

Table 1: Isolated, initial, medial, and final forms of 4 different letters from the Arabic alphabet.

| Letter | Isolated Form | Initial Position | Medial Position | Final Position |
|---|---|---|---|---|
| bā' | ب | بـ | ـبـ | ـب |
| tā' | ت | تـ | ـتـ | ـت |
| lām | ل | لـ | ـلـ | ـل |
| mīm | م | مـ | ـمـ | ـم |

post-Ottoman period in the area we know today as the Middle East (which includes Egypt, but not North Africa).

A few challenges related to OCR/HTR of the Arabic script include the following:

1. The shape of each character of the Arabic script depends upon its contextual position within a word. Many characters have four different shapes depending upon whether they are in their isolated, initial (at the beginning of a word), medial (in the middle of a word), or final forms (at the end of a word). This poses a challenge for the OCR/HTR system that has to recognize all the different forms of the same character. Table 1 shows an example of 4 Arabic characters and their different forms.

2. Additional symbols in Arabic script include ijam, which are dots present above or below a character. Two characters may have the same basic shape but different numbers of dots to tell them apart. For example, the ijam or dots distinguish the bā' (one dot below) from tā' (two dots above), as shown in Table 1. An HTR system may misclassify a character because of the ijam.

3. Arabic language also has diacritics, which include ijam and tashkil. Tashkils are also called harakat. Harakat are short vowel marks in Arabic and are used to indicate the pronunciation of words. They are optional symbols and may not be present in the script. For example, the phrase without diacritics: مجموعة بيانات محرف رائعة turns into مَجْمُوعَةُ بَيَانَاتٍ مُحَرَفٍ رَائِعَةٌ with diacritcs. The diacritics in the later phrase are the accent marks above or below the characters. In the context of an HTR system, the diacritics can get mislabeled for ijam and vice versa. They also increase the size of the character set that the system has to deal with.

## 3 Publicly available Arabic resources

The number of publicly available Arabic datasets is less than those available for languages written in the Latin script. Many Arabic datasets were tailored for specialized tasks, e.g., BADAM for baseline detection [14], HADARA80P for word spotting [26], AHDB for detecting and recognizing numbers on legal checks [4], and WAHD for writer identification [3]. In our subsequent discussion, we focus on offline, handwritten datasets for Arabic HTR, where ground truths for text are available. We may classify offline handwritten HTR datasets into two categories:

**Category 1 (scribed)** In this category, handwritten samples are obtained under controlled conditions by requesting scribes to copy paragraphs or lines of text provided to them. Such a scenario not only offers the flexibility of choosing text and frequency of words to transcribe, but also allows a researcher to choose writing styles, quality/texture of the paper, the writing implement such as pen or pencil, and, scanning/lighting conditions. Moreover, the ground truth is predetermined with the caveat that the handwritten pages have to be manually verified to ensure that a writer has not made any mistakes. The widely used IAM dataset of handwritten English sentences [19] falls into this category. The same goes for the French RIMES dataset [11] and its latest update [10]. Such datasets are better suited for the HTR of contemporary documents.

The earlier Arabic handwritten OCR datasets in this category consisted of word images and their corresponding transcriptions. Examples include the IFN/ENIT dataset [30], which has 26,549 images,

compiled from a vocabulary of 937 Tunisian town names. Similarly, the IFN/Farsi [23] has 7,271 word images of the names of Iranian cities and provinces.

To the best of our knowledge, the KHATT dataset [18] is the first Arabic dataset with paragraph-level handwriting and corresponding ground truth. 1,000 writers filled out forms by copying preprinted text on the same page. Paragraphs were later segmented automatically and manually verified. The MADCAT Phase 1–3 training sets [15] consist of an overall 42,047 scanned handwritten page images. Writers were asked to copy documents by hand using various writing styles (fast, normal, and careful), on lined or unlined paper using a pen or pencil. However, this dataset is not freely available to the public, which restricts its use.

**Category 2 (original)** This category comprises scanned genuine handwritten documents that have been annotated and transcribed by individuals fluent in a respective language. For historical manuscripts, the expertise of historians or linguists may be required. The well-known ICDAR 2017 HTR competition dataset [32] of early modern German language from the READ project belongs to this group with page-level transcriptions (instead of line-level) of more than 10,000 images. Our proposed Muharaf dataset also falls into this category, comprising a collection of historic manuscripts that primarily range from the late 19th to the mid-20th century.

RASM [6] and RASAM [33] are two Arabic datasets of scanned original historic manuscripts. Both datasets have annotated text regions and text lines along with their corresponding transcriptions. A more recent dataset is the Historic Arabic HTR dataset [24] with a collection of 40 pages and their corresponding page-level ground truths. Table 2 summarizes key Arabic HTR datasets that we are aware of.

Table 3 shows the line-level statistics of the IAM English dataset and other publicly available Arabic datasets containing line-level text annotations. Out of the Arabic datasets that are publicly accessible, it is evident that Muharaf contains the largest number of annotated text lines.

**Contrasting features of Muharaf and other available Arabic datasets** The existing Arabic datasets are a valuable resource for the research community. Muharaf supplements them with its own unique features and characteristics, discussed below:

1. As shown in Table 3, Muharaf has the largest number of line images as compared to IAM, RASAM, RASM, and KHATT.

2. KHATT and MADCAT datasets are category 1 (scribed) datasets, where writers were given text to write under controlled experimental conditions. Muharaf is a category 2 (original) dataset, where original historic manuscripts were scanned and transcribed.

3. RASAM and RASM are two Arabic datasets of scanned original historic manuscripts from category 2. The main difference between these two datasets and Muharaf lies in the Arabic script with which they were written (see points 4 and 5 below). Another difference is that RASAM and RASM handwritten pages belong to books, with calligraphic handwriting that is very neat and uniform across pages, written in straight horizontal lines. They were written by scholars in their respective fields. Muharaf includes informal/personal styles of writing, which were very common in the 19th and 20th centuries. The samples vary from very neat to barely legible writing. The handwriting samples of the same individual can be different over different documents or letters. Moreover, the text lines can be slanted upward or downward instead of horizontally straight lines.

4. RASAM has three types of manuscripts from the 10th century. They are scanned pages of books, which were written in the "Meghrebi script" also known as the "Round script" [33]. As the name suggests, this script has very rounded shaped letters. In contrast, Muharaf's documents are mostly Ruq'ah script, which is used for everyday or casual writing. It is composed of straight, short lines, and simple curves.

5. RASM has 4 different types of manuscripts of scientific writings from the 8th century to the 19th century. Muharaf has images from 50 different collections, each collection having one or more writers. Like RASAM, RASM's handwriting styles are calligraphic, very neat, and uniform across all pages as opposed to Muharaf, where the writer may not have very careful or readable handwriting.

Table 2: An overview of key Arabic HTR datasets.

| Dataset | Category | Annotated Text Lines | Total Writers | Vocabulary/Composition |
|---|---|---|---|---|
| IFN/ENIT (2002) [30] | 1 | ✗ | 411 | 937 Tunisian town names. 26,549 word images. |
| AHDB (2002) [4] | 1 | ✗ | 100 | Numbers and phrases used to express numbers on legal checks, 20 most frequently used Arabic words, free hand paragraphs. |
| IFN/FARSI (2008) [23] | 1 | ✗ | 600 | 1,080 Iranian city/province names. 7,271 word images. |
| KHATT (2012) [18] | 1 | ✓ | 1,000 | 2,000 unique text + 2,000 similar text paragraphs. 1,000 writers. Paragraphs segmented automatically and text corrected manually. |
| AHTID/MW (2012) [20] | 1 | ✓ | 53 | Open vocabulary. 3,710 annotated lines. 22,896 words. |
| MADCAT (2012) [15] | 1 | ✓ | 311$^a$ | Document source: weblogs, newswires, and newsgroups. 42,047 page images written by scribes under controlled conditions. |
| HADARA80P (2014) [26] | 2 | ✗ | 1$^b$ | 80 pages scanned from the Taaun book (1430 AD). Annotation of pages, text blocks, and words. Ground truth for 16,720 individual words. |
| VML-HD (2017) [13] | 2 | ✗ | - | 680 pages from historic manuscripts with dates ranging 1088–1451. 159,149 word annotations. |
| RASM (2018) [6] | 2 | ✓ | - | 120 historic scientific manuscript pages. 2,613 annotated text lines. Text regions segmented. |
| RASAM (2021) [33] | 2 | ✓ | - | 300 historic Maghrebi script manuscript pages from 10th century. 7,540 annotated text lines. Text regions segmented. |
| Historical Arabic HTR (2024) [24] | 2 | ✗ | - | 40 pages from 8 different historical books. Transcriptions at the page level. |
| Muharaf (this paper) | 2 | ✓ | - | 1,644 (1,216 public, 428 restricted) pages of historic manuscripts from 1800–2018. Text regions segmented. 36,311 (24,495 public, 11,816 restricted) annotated text lines. |

$^a$ From unique subject IDs in scribe_demographic file.
$^b$ Based on text-independent features for handwriting analysis [27].
- Total number of writers is unspecified or unknown due to the nature of the dataset.

Table 3: A comparison of various HTR datasets in terms of total pages, text regions, and total lines.

| Dataset | Page Count | Text Regions | Line Count |
|---------|-----------|--------------|------------|
| IAM [19] | 1,539 | 1,539 | 13,353 |
| RASAM [33] | 300 | 676 | 7,540 |
| RASM [6] | 120 | 132 | 2,613 |
| KHATT [18] | 4,000$^a$ | 4,000$^a$ | 13,435 |
| Muharaf-public | 1,216 | 3,479$^b$ | 24,495 |
| Muharaf-restricted | 428 | 1,388$^b$ | 11,816 |
| Muharaf | 1,644 | 4,867$^b$ | 36,311 |

$^a$ Includes fixed and unique text paragraphs.
$^b$ Includes main text regions, headings, and floating text regions.

The OCR of text from different Arabic datasets like RASAM and RASM has its own challenges and is by no means an easy feat. We intend to supplement the existing Arabic datasets with a variety of handwritten images with the goal of digitizing handwritten documents from the late 19th century to the mid-20th century.

## 4    Data collection process

We developed an image-labeling software named ScribeArabic[2] specialized for annotating and transcribing Arabic page images of the Muharaf dataset. This software allows a user to annotate text lines in a browser window and transcribe them in a panel next to it. A separate module has the option to label various page elements. A screenshot of this software is shown in Figure 2. We have made the source code for ScribeArabic publicly available (see Section D of the supplementary material for all repository links).

The following steps are involved in labeling a page image using ScribeArabic:

1. Line annotation: Marking a polygonal boundary around each text line.
2. Line transcription: Entering the ground truth transcription for each annotated text line.
3. Defining page elements (if needed): Marking, labeling, and tagging basic page elements such as headings, page numbers, floating text, and graphics.
4. Quality assurance (QA): Verifying that the labeling of a page image is correct.

Transcribing historic handwritten Arabic manuscripts primarily requires experts in archival Arabic, though Steps 1 and 3 permit less specialized involvement. Step 1 for line annotations can be performed by non-Arabic speakers with basic knowledge of the Arabic script. Step 3 for tagging basic page elements generally does not require any Arabic knowledge. However, only an Arabic expert can do the transcriptions and the QA steps.

We assembled a team of expert Arabic speakers to transcribe historic Arabic manuscripts with technical support from machine learning researchers. The text lines of the first 180 images of our curated dataset were annotated by non-Arabic speakers using the ScribeArabic software. A Lebanese Arabic history professor then manually entered the transcriptions into an Excel sheet. We chose Excel for entering transcriptions because initially ScribeArabic supported only line annotations. For the next 1,400+ images, we used our upgraded ScribeArabic software to allow the direct input of transcriptions in a browser window. The annotations and transcriptions for these 1,400+ images were performed by two native Lebanese Arabic speakers adept at archival Arabic. Their transcriptions were checked by a third Lebanese Arabic expert who was also a historian. Section F.1 on page 25 has more details of the qualifications of the annotation and transcription team.

Besides the ScribeArabic software, we employed deep learning to speed up the data collection process. After labeling the first 500+ page images, we trained the SFR system to do a full-page HTR. For transcribing subsequent images, we provided the preliminary line annotation and transcription results

---

[2]For code and a link to demo, please see `https://github.com/MehreenMehreen/ScribeArabic`

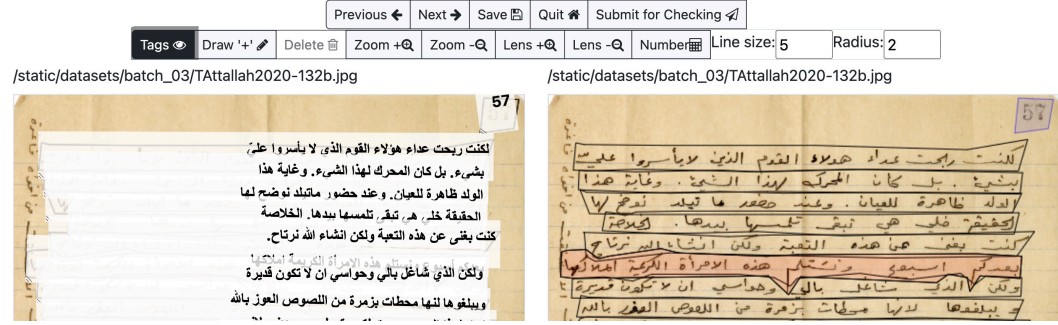

Figure 2: A screenshot showing the graphical user interface of the ScribeArabic software used for labeling page images.

from SFR to the transcribers for manual correction. This streamlined into an iterative process of training SFR with more data and manually correcting the line annotation and transcription on unseen images. The HTR system showed considerable performance improvement over time as we acquired more data. Section 6 has more details of these experiments.

## 5    Muharaf file formats and features

The ground truths for the Muharaf dataset are provided using the page analysis and ground-truth elements (PAGE) XML format [29] and our own Javascript object notation (JSON) file format. The PAGE-XML format is supported for compatibility with Pattern Recognition and Image Analysis (PRImA) Research Lab's Aletheia tool [7] that allows users to open, view, and edit the annotations, transcriptions, and page elements of a document image. They can also view the images using the PRImA Research Lab's PAGE-viewer [28]. We briefly explain the file formats next.

### 5.1    PAGE-XML format for OCR/HTR datasets

The PAGE-XML format is an XML-based page image representation framework introduced by the PRImA Research Lab [31]. This format incorporates various image characteristics as well as the information on page layout and its contents at various levels of granularity. The official, full description of the XML schema is hosted at `https://www.primaresearch.org`. A conceptual class diagram of the subset of the PAGE-XML hierarchy used in the Muharaf dataset is shown in Figure 3. Where applicable, the following labeled regions are present in the PAGE-XML file of each image (see Section D.3 of the supplementary material for illustrative examples).

- Paragraph regions: Main body of text on the page.
- Floating regions: Regions of text outside the normal flow of text. Examples include footer and margin texts.
- Graphics regions: These may include stamps, letterheads, and logos. PAGE-XML format allows these regions to contain text lines.
- Page number regions: Regions containing page numbers.
- Signature regions: Page areas containing names and signatures.

While a large majority of lines in our dataset are handwritten cursive, there are occasional printed characters found on letterheads, logos, or stamps. We have tagged any instance of this printed text as either printed-regular, printed-bold, or printed-italics. We also made the following annotation rules:

- Most of the crossed-out text is annotated, although its transcription is not included.
- We did not manually mark the baseline of the text, but algorithmically computed a mid-line passing through the polygon enclosing the text line. This appears in the XML file under the <UserDefined> tag. This line can be used to determine the orientation of the line and reading direction, which is right-to-left for Arabic and left-to-right for English.

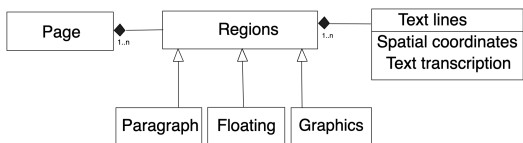

Figure 3: A class diagram of the hierarchy defined by PAGE-XML format. The black diamond indicates the composition relation between two classes. The arrows indicate inheritance. Note: We are using only a subset of the PAGE schema for our dataset.

## 5.2 JSON file formats used by the ScribeArabic software

The JSON format is a simple data-structure dictionary with key–value pairs used to represent entities. Our main data collection process involves the use of ScribeArabic that natively supports the JSON format. We provide two JSON files of different types for each image. The first type has only the line coordinates and their corresponding transcriptions in the "coord" and "text" keys, respectively. These files have the "_annotate.json" suffix and are intended for researchers working only on text-line segmentation and line-level HTR/OCR.

For each image, we also provide a second type of JSON file with the "_tagged.json" suffix. It contains the labeling of various page elements along with individual text-line annotations and transcriptions. Its format is described in detail in Section A of the supplementary material. In our source code, we provide a script to convert the second type of JSON files to the PAGE-XML format. We have also included a custom viewer that reads the JSON file and renders the various page elements of the corresponding image in different colors. Users can use the viewer to browse through images in a directory (Section D of the supplementary material has more details).

## 5.3 Additional characteristics of the Muharaf dataset

Each page image of the Muharaf dataset is part of a specific collection from either the archives of Phoenix Center for Lebanese Studies at USEK or KCLDS at NC State. In the digital archives, a librarian or archivist places all the images of document manuscripts from the same writer, period, or category in a collection. The filenames of all images from the same collection share a prefix that uniquely defines the collection.

A list of 50 collections in the Muharaf dataset and their associated characteristics is presented in Section B of the supplementary material. This list includes an approximate period (if known) for a collection, the total number of image files in a collection, and the top four keywords that describe the collection. The keywords were generated by querying OpenAI's GPT 3.5 APIs [25] with the corresponding Arabic text transcription. The steps and prompts are provided in Section C of the supplementary material.

## 5.4 Line images and their text transcriptions

As handwritten text lines can be slanted or curved, we used the line warping code of SFR [35] to convert them to straight line images. The Muharaf dataset includes a separate directory with text-line images stored in the PNG format and their corresponding text transcriptions stored in the plain text format. This directory is enclosed for the convenience of researchers experimenting with line-level OCR/HTR. They can use this portion of the dataset without having to apply any processing for extracting them from the raw page images or warping them to a straight horizontal image grid. All line images have a height of 60 pixels and a width ranging from 60–2,400 pixels (with an average of 576 pixels).

Figure 4 compares the distributions of image widths among Muharaf and other datasets. The plots reveal that the widths of line images in the IAM dataset are more normally distributed given that the sentences in the dataset were manually selected. The Muharaf, RASAM, and RASM datasets, in contrast, are based on actual documents and all of them have fewer lines that are more than 1,000 pixels wide. Nevertheless, Muharaf has comparatively more line images of shorter widths.

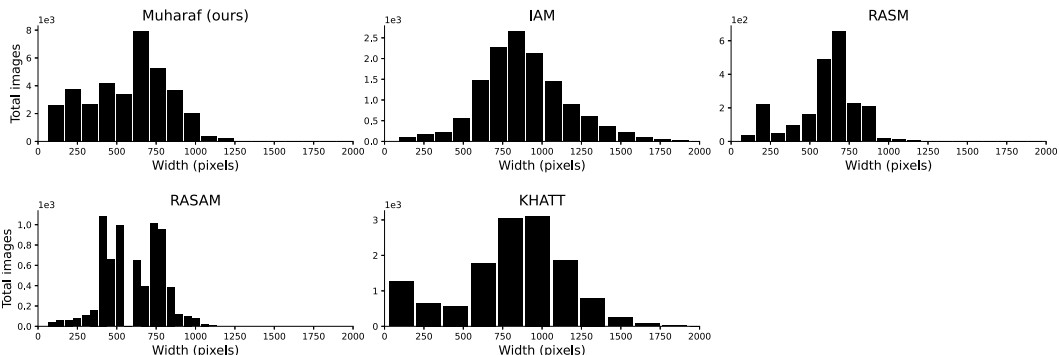

Figure 4: The histograms of the widths of line images for various datasets. Line images were warped to a height of 60 pixels while maintaining the aspect ratio.

## 6 Baseline HTR results and discussion

The SFR [35] system can be trained to do a full-page HTR. It comprises three CNN-based networks:

1. Start of line (SOL) network: A CNN for detecting the coordinates of the start of each text line of a page image.

2. Line follower (LF) network: A CNN that tracks and follows the trajectory of a text line. Starting from the SOL position, this network iteratively predicts the next coordinates and orientation of the text line given its previous coordinates and orientation within a small viewing window.

3. Handwriting recognition (HW) network: A CNN–Bidirectional long short-term memory (BLSTM) network trained using connectionist temporal classification (CTC) loss [9].

Each network in the SFR system can be trained independently, even in a low-resource environment (described in Section 6.1). We used the Muharaf dataset to train the SFR system for full page HTR using a split of 1,500, 50, and 96 images for training, validation, and testing, respectively. The metrics for evaluation are the character error rate (CER) and word error rate (WER), both computed using Levenshtein distance [16] normalized by the length of the string representing the ground truth. The experiments were repeated three times using a different random split of the training, validation, and test sets. Table 4 shows the statistical results of the three experiments in terms of the sample mean and standard deviation. Table 4 also reports the statistical results of training on the public part of the Muharaf dataset. The line-level CER and WER reflect the performance of the HW network on individual pre-segmented text lines extracted using the spatial coordinates of the ground truth annotations. Both error rates are higher than the page-level CER and WER due to the presence of many lines with isolated numbers or single words. If the system makes a mistake on these lines, the CER/WER for these lines can jump to as high as 1.0, contributing to a higher average error rate.

Table 4: HTR results repeated over three random splits of Muharaf. Both page level and line level results are included.

| Dataset | Split (Train, Validate, Test) | Level | CER | WER |
|---|---|---|---|---|
| Muharaf-public | $(1100, 50, 66)$ | Page | $0.157 \pm 0.008$ | $0.398 \pm 0.007$ |
| | | Line | $0.181 \pm 0.009$ | $0.430 \pm 0.011$ |
| Muharaf | $(1500, 50, 96)$ | Page | $0.134 \pm 0.007$ | $0.353 \pm 0.012$ |
| | | Line | $0.149 \pm 0.004$ | $0.380 \pm 0.004$ |

### 6.1 Evolution of HTR performance as more data were collected

We continued to train SFR while the data collection process was going on. Figure 5 illustrates how the system evolved. The plots show the CER and WER over the course of 15 trials. Each trial was repeated three or four times and the sample mean and median statistics are reported. The train, validation, and test sets comprised images for which we had the ground truths available at that time, and hence, the total number of images in the test set of each trial varies. The CER plot reveals that after trial 5 (with 500 training images) the error rate dropped below 20%. We ran the first 11 trials either on Ubuntu 22.04.4 LTS with two Nvidia cards (RTX 3060 12 GB and RTX 2080 8 GB) or on Ubuntu 20.04.6 LTS with a single RTX 4090 24 GB card. Trials 12–14 were run on the NC State's high-performance computing (HPC) cluster.

## 7 Conclusion and Future Work

In this paper, we have introduced a new machine learning dataset of historic handwritten Arabic manuscript images with their annotations and transcriptions. This dataset is a rich collection of a wide range of images with different characteristics. It includes diverse writing styles on a variety of paper backgrounds, ranging from clear handwriting to instances of torn paper or ink bleeds. There are personal notes, diary pages, legal correspondences, financial records, church records, etc. within this dataset, each telling stories of the past and offering a wealth of valuable information. The Muharaf dataset can be used to train a wide variety of systems such as HTR, text-line segmentation, layout detection, and writer identification.

To the best of our knowledge, Muharaf is the largest publicly available Arabic dataset comprising fully annotated and transcribed historical manuscript pages at the text-line level. However, this collection is not devoid of limitations. The process of identifying all the writers and the exact timeline for each document will continue after the initial release of the dataset. For documents where the writer's information cannot be extracted, e.g., the case where a scribe penned a legal document or a church record, one may want to define categories of different writers and writing styles. Another area of interest is the use of Muharaf transcriptions for the extraction of linguistic knowledge and the identification of the colloquial form of the Arabic language used in a particular period. Language models based on this information may improve the performance of the HTR system. We invite researchers in the document analysis and OCR community to utilize this dataset and advance the state of the art.

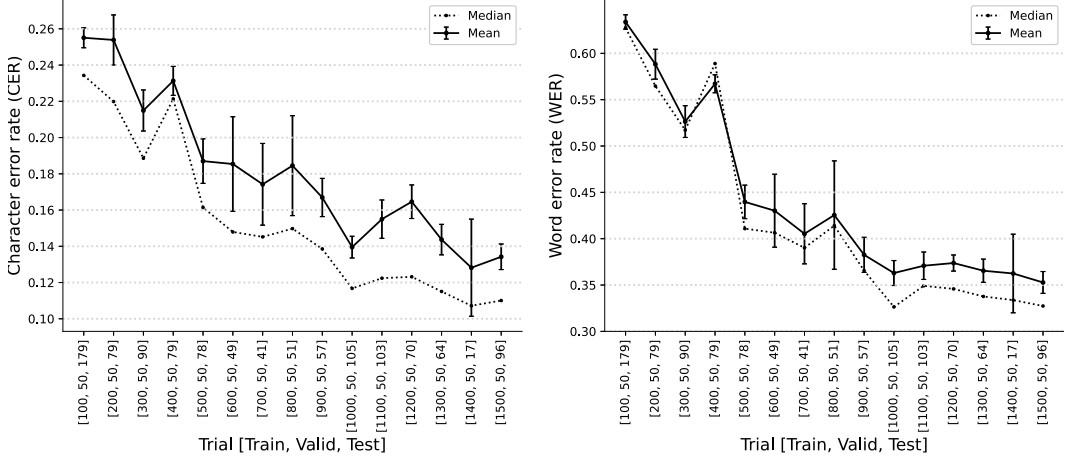

Figure 5: The evolution of the page-level error rates for SFR as we collected more data for creating the Muharaf dataset. The horizontal axis shows the numbers of images in the train, validation, and test sets. The results are averaged over four different random splits for trials 1–8 and three different random splits for trials 9–15.

## Acknowledgments and Disclosure of Funding

We thank Stephen Randall Filios from Family Search for initiating discussions and providing feedback on tagging page elements in document images. We thank Elham Abdallah who is the Assistant University Librarian at USEK for providing support and coordinating work between NC State and USEK.

We acknowledge the computing resources provided by North Carolina State University High-Performance Computing Services Core Facility (RRID:SCR_022168). We also thank Andrew Petersen for his assistance and technical guidance on running jobs on the HPC.

This work was supported in part by the National Endowment for the Humanities (FAIN: ZPA-283823-22), Family Search, and the ECE Undergraduate Research Program at NC State.

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
