# A    JSON file format used in the Muharaf dataset

Each image in the Muharaf dataset comes with two corresponding JSON files. One file has the suffix "annotate" and the other has the suffix"tagged". The two formats are almost identical, except for the addition of tags in the later one. From both JSON files, the transcription and polygonal coordinates of each line can be retrieved from the keys with the prefix "line_". The tags of the "page" JSON format specify whether the marked polygonal area is a text line or region. Additional tags specify whether it is graphics, logo, letterhead, or stamp. The detailed JSON schema is given below:

```
{"$schema": "https://json-schema.org/draft/2020-12/schema",
 "title": "Page",
 "type": "object",
 "properties": {
   "transcriber": {
     "type": "string",
     "description": "Name of the transcriber of this page"
   },
   "taggingBy": {
     "type": "string",
     "description": "Name of the person who tagged page elements
                     (if applicable)"
   },
   "writer": {
     "type": "string",
     "description": "Writer's name if known"
   },
   "comment": {
     "type": "string",
     "description": "Any comment on the page image"
   },
   { "type": "object",
     "patternProperties": {
     "^line_": {
       "type": "object",
       description: "Each polygonal line or region in page image.
                     The tags specify its type"
       "properties": {
         "text": {
           "type": "string",
           "description": "Line transcription"
         },
         "coord": {
           "type": "array"
           "description": "Polygonal coordinates of text line as a flat
                           array [x1, y1, x2, y2, ...]"
         },
         "tags": {
           "type": "object",
           "description": "Tags for additional page elements.
                           Not required if only annotations and
                           transcriptions are needed."
           "properties": {
             "Region_paragraph": {
               "type": "integer",
               "enum": [0, 1]
               },
             "Region_heading": : {
               "type": "integer",
               "enum": [0, 1]
```

```
        },
    "Region_logo": : {
      "type": "integer",
      "enum": [0, 1]
    },
    "Region_letterhead": : {
      "type": "integer",
      "enum": [0, 1]
    },
    "Region_floating": : {
      "type": "integer",
      "enum": [0, 1]
    },
    "Region_pageNo": : {
      "type": "integer",
      "enum": [0, 1]
    },
    "Region_signature": : {
      "type": "integer",
      "enum": [0, 1]
    },
    "Region_graphic": : {
      "type": "integer",
      "enum": [0, 1]
    },
    "Region_decoration": {
      "type": "integer",
      "enum": [0, 1]
    },
    "Region_noise": : {
      "type": "integer",
      "enum": [0, 1]
    },
    "Region_separator": : {
      "type": "integer",
      "enum": [0, 1]
    },
    "Region_linedrawing": : {
      "type": "integer",
      "enum": [0, 1]
    },
    "Region_stamp": : {
      "type": "integer",
      "enum": [0, 1]
    },
    "Archivist_Tag": : {
      "type": "integer",
      "enum": [0, 1]
    },
    "Printed_Regular": : {
      "type": "integer",
      "enum": [0, 1]
    },
    "Printed_Italics": : {
      "type": "integer",
      "enum": [0, 1]
    },
    "Printed_Bold": : {
      "type": "integer",
```

```
            "enum": [0, 1]
            },
        "Orient_top_bottom": : {
            "type": "integer",
            "enum": [0, 1]
            },
        "Orient_bottom_top": : {
            "type": "integer",
            "enum": [0, 1]
            }
        }
      }
     }
    }
   }
  }
 }
}
```

## B  Collections in the Muharaf dataset

The various collections included in the Muharaf dataset and their characteristics are listed in Table 5. Section C describes the procedure for generating the keywords for each collection. If a timeline was unavailable from the archival collection, we assigned it to a broad period. For example, for an individual's collection, we designated the timeline as spanning from the year of birth to the year of death. Future work can continue narrowing it down. The keywords were generated by using OpenAI's GPT 3.5 APIs [25] (details in Section C).

Table 5: Collections in the Muharaf dataset. Each image's filename starts with the string shown in the "Prefix" column. The time period is listed if known.

| No. | Prefix | Total Files | Name | Approximate Time Period | Keywords |
| --- | --- | --- | --- | --- | --- |
| 1 | 00000 | 94 | USEK, Phoenix Center, Patriarche S. Aouad and B. Massad Collection | 1683–1756, 1806–1890 | church administration, legal, financial, official correspondence |
| 2 | 158 | 35 | USEK, Phoenix Center, Al Moutasarrifiya Collection | 1861–1918 | official correspondence, church administration, historical account, legal |
| 3 | 159 | 5 | USEK, Phoenix Center, Al Moutasarrifiya Collection | 1861–1918 | official correspondence, government, politics, legal |
| 4 | 160 | 13 | USEK, Phoenix Center, Al Moutasarrifiya Collection | 1861–1918 | official correspondence, politics, historical account, personal letter |
| 5 | 2015 | 10 | Khayrallah Center, El-Khouri Letters Collection | | personal letter, church administration, historical account, official correspondence |

| No. | Prefix | Total Files | Name | Approximate Time Period | Keywords |
|---|---|---|---|---|---|
| 6 | 0105-12P | 11 | USEK, Phoenix Center, PV Collection | | church administration, financial, legal, historical account |
| 7 | 1940Jbeil | 19 | USEK, Phoenix Center, PV Collection | | church administration, financial, legal, historical account |
| 8 | 26025 | 3 | USEK, Phoenix Center, Al Moutasarrifiya Collection | 1861–1918 | legal, deed, official correspondence, marriage record |
| 9 | 26066 | 3 | USEK, Phoenix Center, Al Moutasarrifiya Collection | 1861–1918 | legal, financial, official correspondence, legal agreement |
| 10 | 27404 | 15 | USEK, Phoenix Center, Al Moutasarrifiya Collection | 1861–1918 | historical account, economy, financial, government |
| 11 | AF | 25 | USEK, Phoenix Center, Amin Farhat Collection | 1916–1941 | financial, personal letter, legal, official correspondence |
| 12 | AJN | 13 | USEK, Phoenix Center, Joseph Nehme Collection | 1910–1994 | historical account, personal letter, spiritual, biography |
| 13 | AP | 29 | USEK, Phoenix Center, Lebanese Maronite Missionaries Collection | | church administration, legal, official correspondence, analysis |
| 14 | AR | 159 | Khayrallah Center, Ameen Rihani Collection | 1876–1940 | historical account, official correspondence, personal letter, biography |
| 15 | AnF | 8 | USEK, Phoenix Center, Antoine Ferjane Collection | 1906–1931 | church administration, historical account, official correspondence, personal letter |
| 16 | BEK | 15 | USEK, Phoenix Center, Bechara El Khoury Collection | 1885–1968 | poem, official correspondence, spiritual, legal |
| 17 | Baddour | 3 | Khayrallah Center. Baddour Collection | | poem, spiritual, historical account, legal |
| 18 | Church | 20 | USEK, Phoenix Center, Churchrecord Collection | | marriage record, church administration, baptism record, historical account |
| 19 | EAC | 21 | USEK, Phoenix Center, Elias Abou Shabake Collection | 1926–1928, 1956–1968 | poem, historical account, spiritual, biography |

Table 5 continued…

| No. | Prefix | Total Files | Name | Approximate Time Period | Keywords |
|-----|--------|-------------|------|-------------------------|----------|
| 20 | EAK | 98 | USEK, Phoenix Center, Bishop Abdallah Khoury Collection | 1872–1949 | church administration, official correspondence, personal letter, spiritual |
| 21 | ES | 18 | USEK, Phoenix Center, President Elias Sarkis Collection | 1924–1985 | government, historical account, politics, economy |
| 22 | FC | 21 | USEK, Phoenix Center, President Fouad Chehab Collection | | historical account, personal letter, financial, biography |
| 23 | HM | 41 | USEK, Phoenix Center, Hanna Moussa Collection | 1959–1970 | poem, spiritual, historical account, biography |
| 24 | JEH | 34 | USEK, Phoenix Center, Joseph El Hachem Collection | 1925–2018 | historical account, poem, personal letter, spiritual |
| 25 | JPK | 24 | USEK, Phoenix Center, Jean Philipp Kmeid Collection | | biography, poem, analysis, historical account |
| 26 | JoM | 98 | USEK, Phoenix Center, Joseph Mikhail Collection | | church administration, legal, official correspondence, financial |
| 27 | KEllis | 30 | Khayrallah Center, Kellis Collection | 1946–1987 | personal letter, historical account, biography, official correspondence |
| 28 | KJoseph | 22 | Khayrallah Center, KJoseph Collection | | official correspondence, financial, legal, historical account |
| 29 | ME | 12 | USEK, Phoenix Center, Mansour Eid Collection | 1944–2013 | analysis, politics, economy, historical account |
| 30 | MG | 33 | USEK, Phoenix Center, Maurice Gemayel Collection | | government, official correspondence, economy, personal letter |
| 31 | MH | 44 | USEK, Phoenix Center, Father Michel Hayek Collection | 1928–2005 | historical account, spiritual, politics, analysis |
| 32 | MISC_R | 13 | USEK, Phoenix Center, Five Registers Collection | 1906–1911, 1953–1954 | church administration, baptism record, marriage record, historical account |
| 33 | MK | 25 | USEK, Phoenix Center, Michel Kahwaji Collection | 1912–2011 | poem, biography, historical account, speech |

| No. | Prefix | Total Files | Name | Approximate Time Period | Keywords |
|-----|--------|-------------|------|------------------------|----------|
| 34 | MUKF | 53 | USEK, Phoenix Center, Papers of Kfarchima Municipality Collection | | financial, official correspondence, legal, government |
| 35 | MeM | 8 | USEK, Phoenix Center, Meneem Meneem Collection | 1837–1855, 1967 | legal, church administration, biography, financial |
| 36 | Misc | 25 | USEK, Phoenix Center, Five Registers Collection | 1906–1911, 1953–1954 | baptism record, marriage record, church administration, historical account |
| 37 | Nasrallah | 48 | Khayrallah Center, Narallah Collection | | historical account, biography, analysis, official correspondence |
| 38 | OLM | 86 | USEK, Phoenix Center, Five Registers Collection | 1955–1963 | legal, church administration, financial, deed |
| 39 | Oussani | 21 | Khayrallah Center. Oussani Collection (Ottomon) | | historical account, travel log, government, official correspondence |
| 40 | QAJE | 19 | USEK, Phoenix Center, Jezzin Collection | 1875–1950 | legal, official correspondence, financial, historical account |
| 41 | QBat | 40 | USEK, Phoenix Center, Al Batroun Collection | 1875–1950 | legal, financial, official correspondence, deed |
| 42 | SB | 13 | USEK, Phoenix Center, Sami Behnan Collection | | personal letter, spiritual, church administration, historical account |
| 43 | ST | 48 | USEK, Phoenix Center, Salah Tizani Collection | 1959–2000 | dialog, official correspondence, historical account, personal letter |
| 44 | TAttallah | 30 | Khayrallah Center, TAttallah Collection | | official correspondence, personal letter, financial, historical account |
| 45 | TB | 6 | USEK, Phoenix Center, Toufic El Basha Collection | 1924–2005 | official correspondence, personal letter, biography, financial |
| 46 | YAL | 21 | USEK, Phoenix Center, Youssef Abdallah Lahoud Collection | 1800–1924 | personal letter, official correspondence, financial, legal |
| 47 | YFS | 14 | USEK, Phoenix Center, Youssef Fadlallah Salemeh Collection | 1906–2001 | personal letter, travel log, official correspondence, historical account |

Table 5 continued...

| No. | Prefix | Total Files | Name | Approximate Time Period | Keywords |
|---|---|---|---|---|---|
| 48 | YS | 81 | USEK, Phoenix Center, Youssef Hanna El-Sawda Collection | 1887–1969 | historical account, official correspondence, politics, government |
| 49 | kc0061 | 49 | Khayrallah Center, Miguel Saikali Collection | | personal letter, historical account, travel log, spiritual |
| 50 | kc0066 | 68 | Khayrallah Center, KC0066 Diary Collection | | historical account, church administration, biography, travel log |

## C   Procedure for generating keywords for each collection

We employed OpenAI's `gpt-3.5-turbo-0125` model [25] to generate keywords for every collection in the Muharaf dataset. For each page image in a collection, we instructed GPT to generate a list of three keywords based on the Arabic text transcription of that page image. Next, we created a list of keywords and their corresponding frequencies for all pages in that collection. The top 4 occurring keywords present in the collection are listed in Table 5. The prompt used to generate keywords for an individual page image is given below:

```
You will be given Arabic text. Based on the text, assign it the most three
relevant keywords from this list:

1 official-correspondence
2 personal-letter
3 church-administration
4 analysis
5 legal
6 financial
7 marriage-record
8 baptism-record
9 census-record
10 economy
11 administrative
12 politics
13 speech
14 biography
15 government
16 poem
17 dialog
18 historical-account
19 commentary
20 spiritual
21 travel-log
22 deed
23 legal-agreement

Constraints:
1. Output the three keywords and their indices.
2. Keyword indices (1, 2) cannot occur together.
3. An index can only be a number from 1, 2, ...,  23.
4. Give output as JSON format with keys:
```

```
    keywords_list_of_length_3,indices_list_of_length_3
5. Don't output anything additional.
```

Note that the prompt and the keywords present above were manually refined iteratively based on the output of GPT. We generated these keywords for the general interest of the research community who are non-Arabic speakers. However, we do not claim that these summaries and keywords are 100% correct.

## D   Downloading Muharaf data, related software, code, and license

The GitHub repository containing instructions on downloading the Muharaf dataset and links to all related code can be found at `https://github.com/mehreenmehreen/muharaf`. We briefly describe the contents of this repository and the associated licenses next.

### D.1   Dataset download

The Muharaf GitHub repository has a Zenodo link to download the public part of the Muharaf dataset. It has 1,216 images, which are hosted on Zenodo at `https://zenodo.org/records/11492215`. Users have the option to download the following:

- Public part of data files that contain page images and their corresponding annotation files. Both JSON and Page-XML files are included in the annotation files.
- Individual line images extracted from the page images. The line images are available for the public part of the Muharaf dataset.
- Summary and keyword files corresponding to each image of the public part of the Muharaf dataset. The keywords were extracted using the procedure described in Section C.

The restricted part of the Muharaf dataset has 428 images distributed under a proprietary license. The images, annotations, line images, and summaries for the restricted portion of the Muharaf dataset can be obtained by writing to Carlos Younes `carlosyounes@usek.edu.lb` at Phoenix Center for Lebanese Studies, USEK.

### D.2   ScribeArabic annotation software

All the annotations of the Muharaf dataset were created using the ScribeArabic software. The transcriptions for more than 1,400 lines were also entered using ScribeArabic. The ScribeArabic annotation software is a Django app. The source code for this app is hosted on GitHub at `https://github.com/mehreenmehreen/ScribeArabic` and its manual is at `https://github.com/mehreenmehreen/ScribeArabic/blob/main/manual.md`. While ScribeArabic was designed to annotate and transcribe Arabic page images, it can easily be adapted for other languages. It can also be adapted for labeling images for other computer vision applications.

### D.3   PAGE-XML converter and page elements viewer

We include the code for converting ScribeArabic's JSON files to PAGE-XML files using a PAGE-XML converter. Instructions for downloading the code and running it are at `https://github.com/mehreenmehreen/xml_converter`. Users can also download a custom viewer for inspecting the annotated page elements on a document page. The viewer shows the different annotated page elements in different colors. A screenshot of this viewer is shown in Figure 6(a). This app was written in Python using the Tkinter library. Figure 6(b) illustrates examples of various page elements described in Section 5.1.

### D.4   Start, Follow, Read — Arabic

Users can replicate the results of all experiments reported in this paper by downloading the adapted source code of Start, Follow, Read (SFR) [35]. The GitHub repository for the Arabic version is hosted at `https://github.com/mehreenmehreen/start_follow_read_arabic`. This repository contains code and links for the following:

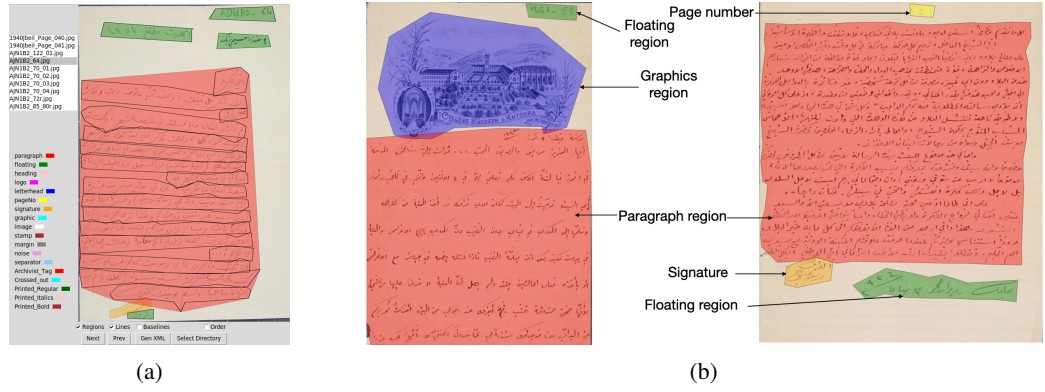

Figure 6: (a) Viewer of the annotated page elements showing floating regions, a paragraph region, and a signature region. Individual text lines are enclosed in black polygonal boundaries. (b) Various examples of different page elements including main paragraph regions, floating regions, a page number region, a signature region, and a graphics region.

- Code for preprocessing the image files and their corresponding JSON files. All data folders and files created after running the preprocessing step can also be downloaded via a provided Zenodo link.
- Code for training the individual SOL, LF, and HW networks.
- Folders for replicating the experiments on the public portion of the Muharaf dataset and the entire dataset. Both folders have three directories, set0, set1, and set2. Each set contains the training, validation, and test splits for running the experiments. Each set also contains the YAML configuration files that specify all the hyperparameters for training SFR-Arabic.
- The trained SFR model weights obtained from training both the Muharaf dataset and its public portion. Users can run the inference code to get the error rates on various image files. Users can also run the full HTR code on a page image and generate JSON files containing the predictions from SFR-Arabic.
- Additional inference results from training the Muharaf dataset and its public portion. There are also additional cross-dataset evaluation results.

### D.5 Pretraining the model

The SOL and LF networks of SFR-Arabic were trained from scratch with random weights. We pretrained the HW model using synthetically generated Arabic lines. The text of the Arabic lines was taken from the Khaleej-2004 newspaper corpus [1] and Watan-2004 corpus [2]. We have provided the details of the pretrained model on our Github webpage with a link to download it.

### D.6 License

We release our data and code under the following licenses:

- The public part of the Muharaf dataset has 1,216 images distributed using the Creative Commons license CC BY-NC-SA 4.0. Users are free to share and adapt the dataset under the terms of attribution, non-commercial use, and share alike, as specified by the Creative Commons license at https://creativecommons.org/licenses/by-nc-sa/4.0/.
- The restricted part of the dataset has 428 images distributed under a proprietary license. It can be downloaded by writing to Carlos Younes carlosyounes@usek.edu.lb at Phoenix Center for Lebanese Studies, USEK. This part of the dataset is distributed under a proprietary license with the condition that it will not be redistributed and only be used for research purposes.
- The source code for the ScribeArabic annotation software and XML converter and viewer are also released under the Creative Commons license CC BY-NC-SA 4.0. Users are free to

share and adapt the dataset under the terms of attribution, non-commercial use, and share alike, as specified by the Creative Commons license at https://creativecommons.org/licenses/by-nc-sa/4.0/.

- The source code for START, FOLLOW, READ — ARABIC is modified from the source code released by the authors of Start, Follow, Read [35]. Their code is free to use for academic and research purposes as given on their GitHub page: https://github.com/cwig/start_follow_read?tab=readme-ov-file. We release the Arabic version under the same license. [3]

## D.7 Ethics statement: privacy, legal, or ethical issues

Muharaf-public has manuscript images from the archives of Phoenix Center for Lebanese Studies at USEK and KCLDS at NC State. These images were already publicly accessible before our proposed redistribution effort. Readers could access the images through the centers' websites or as hard copies through the libraries. The research process described in this manuscript sticks to the ethical standards, and there are also no concerns regarding the leakage of personal information.

Muharaf-restricted is distributed under a restricted license. The limited licensing is due to the proprietary nature of the images, and hence are authorized to use only with permission from the owners. The usage restrictions include that the dataset shall be used solely for research purposes and not be redistributed without permission. We will require researchers to agree to the statement of ethical use of data as a requirement of downloading the restricted part of the dataset.

## E   Additional comparisons and cross-dataset validation results

Table 6 shows the comparison of SFR with a transformer based system TrOCR [17] on the IAM dataset [19]. While the performance of TrOCR is better than SFR, it has a higher computational complexity and requires more resources. Our choice of SFR is based on the goal of developing a system for a low-resource environment. Each network in this system can fit on an 8 GB card (mentioned in Section 6.1), making this system ideal for a low-resource environment. State-of-the-art models like transformers cannot be realistically deployed in a resource-constrained setting. By choosing a more traditional CNN-based network, we are willing to trade a slight reduction in accuracy for a smaller, less resource-intensive model.

Table 7 shows the results of cross-dataset evaluation on the KHATT dataset [18]. When SFR was trained with Muharaf or RASAM+RASM datasets, the accuracy of both trained models was not very good. However, when all three datasets were combined, the system's CER improved by ∼16%, showing the significance of Muharaf as a valuable addition to the current publicly available Arabic datasets. The first row of Table 7 shows the baseline CER of a transformer based system [21] trained on the KHATT dataset itself. We can see that the transformer has significantly large parameters (153.1M) as compared to the HW network of SFR, which has only 18M parameters.

Table 6: Comparison of TrOCR [17] and SFR [35] on the IAM dataset.

| Model | CER (%) | Parameters |
|---|---|---|
| TrOCR$_{SMALL}$ [17] | 4.22 | 62M |
| TrOCR$_{BASE}$ [17] | 3.42 | 334M |
| TrOCR$_{LARGE}$ [17] | 2.89 | 558M |
| SFR [35] | | SOL-9M, |
| (used as baseline for Muharaf) | 6.40 | LF-5M, |
| | | HW-18M |

---

[3]The authors agree to bear all responsibility in case of violation of rights, etc.

Table 7: HTR performance of systems on KHATT dataset trained with different datasets. The test images are the line images from the test set in KHATT's "unique paragraph" directory.

| Model | Training Dataset | CER (%) | Parameters |
|---|---|---|---|
| Transformer with cross-attention [21] | KHATT | 18.45 | 153.1M |
| HW (From SFR) | Muharaf | 38.45 | 18M |
| HW (From SFR) | RASAM + RASM | 41.14 | 18M |
| HW (From SFR) | Muharaf + RASAM + RASM | 24.05 | 18M |

# F  Datasheet for Muharaf Dataset

We organize this section according to the relevant portions of the datasheet for datasets template [8].

## F.1  Motivation

**For what purpose was the dataset created?**  The primary goal of creating this dataset is to train an OCR/HTR system capable of digitizing handwritten Arabic manuscripts and documents in a digital library, archive, or collection, making them accessible and searchable.

**Who created the dataset?**  A majority of the annotations and transcriptions (1400+ images) were completed by Arabic speakers who manage digital archives of Arabic manuscripts and facilitate information access from a large number of Arabic documents. Their exact designations are:

- Carlos Younes: Head of the reference and external relations at Phoenix Center for Lebanese-Historical Archives (PCLS), USEK.
- Georges Habchi: Head of Storage Division at Digital Development Center, USEK.

The transcriptions of both individuals were checked by a full professor of history who is also a historian and the director of USEK Library, Phoenix Center for Lebanese Studies.

Some transcriptions (around 180 images) were completed by:

- Amin Elias: Assistant professor teaching history at the center for Lebanese studies.

These transcriptions were checked by a full professor of history at KCLDS, NC State. Each image of the packaged Muharaf dataset is accompanied by a JSON and XML file. The JSON and XML files contain the names of the annotator and the transcriber of that image. Table 8 summarizes the total number of images transcribed by each team member.

**Who funded the creation of the dataset?**  This work was supported in part by a grant from the National Endowment for the Humanities (NEH), FAIN: ZPA-283823-22. It was also supported in part by Family Search and the Electrical and Computer Engineering Undergraduate Research Program at NC State.

Table 8: Summary of total images transcribed by each member of the transcription team.

| Transcriber | Total images transcribed |
|---|---|
| Amin Elias | 179 |
| Carlos Younes | 663 |
| Georges Habchi | 802 |
| Total | 1644 |

### F.2 Composition

**What do the instances that comprise the dataset represent?** Each instance of the dataset is a document page image. Almost all images are scanned pages of handwritten Arabic, except 21 images that are in handwritten Ottoman Turkish. Three images are scanned typewritten pages. There are different types of individual page images, e.g., personal letters, poems, notes, diary images, legal correspondences, and church records. The timeline for this dataset ranges from the late 19th to the early 21st century.

**How many instances are there in total?** There are:

- 1,644 image files of scanned handwritten Arabic document pages. 1,216 image files are public and 428 files are restricted.
- 36,311 (24,495 public and 11,816 restricted) text lines.

**Does the dataset contain all possible instances or is it a sample of instances from a larger set?** The dataset is complete and contains all possible instances.

**What data does each instance consist of?** Each instance consists of:

- A scanned handwritten Arabic document image (JPEG format). This is for researchers working on full page OCR/HTR.
- A processed line image (PNG format). This is for researchers working on OCR/HTR of text line images.

**Is there a label or target associated with each instance?** The target/label for each page image represents:

- Spatial polygonal coordinates of individual text lines in a document image.
- Transcription of each text line.
- Spatial coordinates of a page element and its type. The type can be:
    - Paragraph region.
    - Floating text region. Any text outside the normal flow of text is labeled as a floating text region.
    - Page number region.
    - Signature region. This region contains names and signatures.
    - Graphics region. This region can contain logos, stamps, or letterhead images. Text lines are also allowed in this region.

Each page image is accompanied by:

- One JSON file with "_annotate" in its filename and containing the annotations and transcriptions of all text lines in the page image.
- One JSON file with "_tagged" suffix containing the annotation and transcriptions of each text line. It also contains the annotation of various page elements on the document image.
- A PAGE-XML file for compatibility with PRImA Research Lab's Aletheia tool [7] and PRImA Research Lab's PAGE-XML viewer [28].
- A plain text file containing an English summary of the ground truth of the document page image.
- A plain text file containing keywords of the document page image in English. The keywords were generated using OpenAI's GPT APIs [25].

For each line image, the label is its transcription, which is contained in a plain text file.

**Is any information missing from individual instances or labels?** A few text lines in the margins, footnotes, or signatures are not annotated and transcribed. Also, some of the page elements like separators or noise are not annotated.

**Are there recommended data splits?** No. We trained the system with three different random splits of training, validation, and test sets (1500, 50, 96). Our GitHub website includes links to download the three different sets of data splits.

**Are there any errors, sources of noise, or redundancies in the dataset?** As images vary in each document, the labeling of page elements such as floating text, heading, and main paragraphs was done using the personal judgment of the annotator. While the transcriptions have gone through a QA round, there are still some minor errors in the dataset:

- In very rare cases, some words may not be transcribed properly.
- The annotations of many text lines were generated automatically, and hence, they are not tight polygons around the line. This implies that many annotations overlap and a bounding polygon may contain some portions of the polygon from the line above or below it.
- Some signatures and floating areas have multiple text lines contained in the same line annotation.

**Is the dataset self-contained, or does it link to or otherwise rely on external resources?** The dataset is self-contained.

**Does the dataset contain data that might be considered confidential?** The dataset does not contain any confidential information. The public portion of the dataset includes page images that are already publicly accessible. For the restricted part of the dataset, the images are proprietary and require permission from the owner for use only in non-commercial research.

**Does the dataset relate to people?** Yes. The dataset includes many document images such as personal letters, diaries, and official records from various personal collections. The documents are from the early 19th century to the early 21st century and do not relate to living people.

**Does the dataset identify any subpopulations?** No

**Is it possible to identify individuals, either directly or indirectly from the dataset?** Yes, for many page images, the writer of the document can be identified.

**Does the dataset contain data that might be considered sensitive in any way?** No

### F.3 Collection process

**How was the data associated with each instance acquired?** An image from a particular archive collection was identified and used for annotation and transcription.

**What mechanisms or procedures were used to collect the data?**

- For 180 images, the text lines were annotated using the ScribeArabic software. The text was entered in an Excel sheet by a history professor and manually verified by another history professor.
- During the data collection process, the available data was used to train the Start, Follow, Read (SFR) system [35]. The annotations and transcriptions of many images were automatically generated using SFR.
- For more than 1,400 images, a team of two Lebanese Arabic speakers who are also archivists used the ScribeArabic software to either:
  - Correct the annotations and transcriptions generated by SFR.
  - Manually annotate and transcribe the text lines on a document page image from scratch.
- The annotations and transcriptions of the 1,400+ images were checked by a native Lebanese Arabic expert who is also a historian.
- A team of non-Arabic speakers annotated the images with various page elements. These page elements were also manually verified.

**Effectiveness of QA process**   To assess the effectiveness of the QA process, we evaluated the CER and WER of several batches of received transcriptions both before and after the QA process. The results are summarized in Table 9. As shown, only minimal changes were made to the transcriptions during the QA phase.

**Over what timeframe was the data collected?**   March 2023–April 2023, July 2023–March 2024.

**Were any ethical review processes conducted?**   No

Table 9: CER and WER of transcriptions of 5 different batches before and after the QA phase.

| Date Batch Received (2024) | CER (%) | WER (%) | Total Images |
|---|---|---|---|
| 12/01/2023 | 0.051 | 0.197 | 53 |
| 12/23/2023 | 0.113 | 0.581 | 55 |
| 01/02/2024 | 0.256 | 0.890 | 83 |
| 01/16/2024 | 0.176 | 0.700 | 98 |
| 01/26/2024 | 0.071 | 0.397 | 67 |

### F.4   Preprocessing/cleaning/labeling

**Was any preprocessing/cleaning/labeling of the data done?**   Yes.

- Some page images were cropped before annotation.
- The folder with line images has text lines along with their corresponding transcriptions. The individual lines were extracted from document images using preprocessing routines of the SFR system.

**Is the software used to preprocess/clean/label the instances available?**   Yes, the software is available.

- The original SFR code is available at: `https://github.com/cwig/start_follow_read`
- The SFR code adapted for Arabic is available at `https://github.com/mehreenmehreen/muharaf` link.

### F.5   Uses

**Has the dataset been used for any tasks already?**   Yes, it has been used to train a full-page HTR system as described in Section 6 of the main paper.

**Is there a repository that links to any or all papers or systems that use the dataset?**   We have set up a GitHub page for this purpose `https://github.com/mehreenmehreen/muharaf`.

**What tasks could the dataset be used for?**   The dataset can be used to:

- Train a system for text line segmentation.
- Train a system for OCR/HTR.
- Train a system for layout detection.
- Train a language model using the ground truth transcriptions.
- Linguists can study the colloquial form of Arabic for various periods.
- Identify various writing styles in a given period.
- Identify various writing styles used to record legal documents or church records.

**Is there anything about the composition of the dataset or the way it was collected and prepro-cessed/cleaned/labeled that might impact future uses?**    No.

## F.6    Distribution

**Will the dataset be distributed to third parties outside of the entity on behalf of which the dataset was created?**    Yes, the dataset will be distributed to researchers.

**How will the dataset be distributed?**    The public portion of the Muharaf dataset is hosted on Zenodo at https://zenodo.org/records/11492215. The related software and source code is hosted on GitHub at https://github.com/mehreenmehreen/muharaf. This GitHub page includes instructions on downloading the restricted portion of Muharaf.

**When will the dataset be distributed?**    The links for downloading Muharaf-public are active now. The restricted part of Muharaf is also available upon request.

**Will the dataset be distributed under a copyright or other intellectual property (IP) license, and/or under applicable terms of use (ToU)?**

- The public part of the dataset has 1,216 images distributed using the Creative Commons license CC BY-NC-SA 4.0. Users are free to share and adapt under the attribution, non-commercial, and share alike terms of the Creative Commons license as given at https://creativecommons.org/licenses/by-nc-sa/4.0/.

- The restricted part of the dataset has 428 images distributed under a proprietary license. It can be downloaded by writing to Carlos Younes carlosyounes@usek.edu.lb at Phoenix Center for Lebanese Studies, USEK. This part of the dataset is distributed under a proprietary license with the condition that it will not be redistributed and only be used for research purposes.

**Have any third parties imposed IP-based or other restrictions on the data associated with the instances?**    The restricted part of the dataset has 428 images distributed under a proprietary license with permission from the owners.

**Do any export controls or other regulatory restrictions apply to the dataset or to individual instances?**    No.

## F.7    Maintenance

**Who is supporting/hosting/maintaining the dataset?**    The dataset is hosted on Zenodo. It will be maintained by the research teams working at Khayrallah Center for Lebanese Diaspora Studies (KCLDS) and Electrical and Computer Engineering Department (ECE) at NC State.

**How can the owner/curator/manager of the dataset be contacted?**    Users can contact the teams managing the dataset by directly opening an issue on the GitHub page: https://github.com/mehreenmehreen/muharaf. They can also contact them using the emails provided on the GitHub page.

**Is there an erratum?**    We'll build an erratum over time as more and more researchers start using Muharaf.

**Will the dataset be updated?**    Yes, transcription errors pointed out by users will be corrected and updated after verification.

**Will older versions of the dataset continue to be supported/hosted/maintained?**    Yes, we plan to host all versions of our dataset on Zenodo.

**If others want to extend/augment/build on/contribute to the dataset, is there a mechanism for them to do so?** We have released the code for the ScribeArabic software, which can be used to annotate and transcribe page images. It can also be used for labeling various page elements. Other researchers can use ScribeArabic to build similar datasets. We welcome any additional contributions and are willing to add them to our dataset after verification.