# OpenReview forum: "Muharaf: Manuscripts of Handwritten Arabic Dataset for Cursive Text Recognition"
_NeurIPS.cc/2024/Datasets_and_Benchmarks_Track — NeurIPS 2024 Track Datasets and Benchmarks Poster_

### Official Review · Reviewer_y5EV · 2024-07-18
**A dataset consisting of over 1,600 historic handwritten Arabic pages**

**Rating:** 6
**Confidence:** 5
**Correctness:** The dataset is constructed in a sound…
**Clarity:** The paper is well written.

**Review:**

- The dataset proposed by this paper aims to advance the state-of-the-art in handwritten text recognition (HTR) for Arabic manuscripts and cursive text in general. It includes diverse handwriting styles and various document types such as personal letters, diaries, notes, poems, church records, and legal correspondences.

- The authors describe the data acquisition pipeline, notable dataset features, and statistics. They also provide a preliminary baseline result achieved by training convolutional neural networks (CNNs) using this dataset.

- The proposed dataset was constructed by Arabic experts and was checked after the preliminary annotation, thus it is high-quality.

- Overall, the Muharaf dataset offers a valuable resource for researchers working on Arabic HTR, text-line segmentation, layout analysis, writer identification, and related tasks.

**Strengths:**

(1) The Muharaf dataset stands out with the largest number of annotated text lines among Arabic Handwritten Text Recognition datasets.

(2) The inclusion of diverse handwriting styles and a wide range of document types makes the Muharaf dataset valuable.

(3) The dataset provides annotations not only for coordinates and texts but also for basic page elements, enhancing its usefulness.

**Additional Feedback:**

None

**Documentation:**

This dataset provides details on data collection and organization, availability and maintenance, and ethical and responsible use.

**Ethics:**

There are no ethical concerns.

**Limitations:**

The authors discuss some of the limitations in Sec. 6.

**Opportunities For Improvement:**

(1) The authors should emphasize the significance of the Muharaf dataset in relation to existing Arabic datasets.

(2) The baseline experiments only focus on a single method proposed six years ago, limiting the exploration of newer approaches.

(3) It is important to mention any licenses associated with the data sources to ensure proper attribution and compliance.

(4) Providing illustrative examples of the annotations of basic page elements would enhance the clarity of the dataset description.

**Relation To Prior Work:**

This paper has discussed how this work differs from previous contributions.

**Summary And Contributions:**

This paper introduces a Manuscripts of Handwritten Arabic dataset, Muharaf. The dataset consists of more than 1,600 historic handwritten page images with spatial polygonal coordinates of its text lines as well as basic page elements. The dataset includes diverse handwriting styles and a wide range of document types. It can be used to train a wide variety of systems such as HTR, text-line segmentation, layout detection, and writer identification.

---

> ### Author Rebuttal · Authors · 2024-08-15
>
> We thank Reviewer y5EV for providing an insightful and helpful review of our paper. We’ll revise our paper in accordance with these comments and feedback. Here is a response to the issues raised in the review.
>
> # Significance of Muharaf in relation to existing Arabic datasets
> We have done a detailed comparison with existing Arabic datasets. Please see the global rebuttal for our response.
>
> # Baseline experiments with Start, Follow, Read
> We agree with the reviewer that the Start, Follow, Read, end-to-end full page handwriting recognition system (SFR) is a traditional OCR system. One of the goals of our work is to enable OCR of languages in a low resource setting. Keeping this goal in view, we chose SFR for carrying out baseline experiments. Each network in this system can fit on an 8GB card (mentioned on Page 2 of Section 1), making this system ideal for a low resource environment. State-of-the-art models like transformers cannot be realistically deployed in a resource constraint setting. The main contribution of our paper is the Muharaf dataset, which we want to make available to the research community. Publishing this dataset will encourage other researchers to engage with it and test the limits of their OCR systems.
>
> We can take the example of a transformer based model like TrOCR, which has 62M, 334M, or 558M parameters depending upon whether it is the small, base, or large model [14]. These models have a CER of 4.22%, 3.42%, and 2.89% respectively on the IAM dataset. In contrast the CNN based SFR model has a CER of 6.4% but comparatively fewer parameters. The SOL, LF, and HW networks of the system have 9M, 5M, and 18M parameters respectively. Moreover, all three of them can be loaded and trained independently. Therefore, by choosing a more traditional CNN-based network, we are willing to trade a slight reduction in accuracy for a smaller, less resource-intensive model. Please see the global PDF for additional results on IAM and KHATT datasets.
>
>
>
> # License of other data sources
> Muharaf’s main data sources are the handwritten images. All the Muharaf-public images are either available in the archives of the Phoenix Center for Lebanese Studies at Holy Spirit University of Kaslik (USEK) or the Khayrallah Center for Lebanese Diaspora Studies (KCLDS) at North Carolina State University (NC State). The images of Muharaf-public are from different collections that are under different licenses, and the donors/owners of these images have granted permission to the respective centers to use them for scientific research. The URLs for the two centers are:
> [Phoenix Center](https://www.usek.edu.lb/phoenix-en/home)
> [Khayrallah Center](https://lebanesestudies.ncsu.edu/archive/)
>
> Muharaf-restricted has proprietary images and hence, requires access permissions from the owners under a restricted license.
>
> # Illustrative examples of basic page elements
> We have described various page elements in Section 4.1 (page 6) of our main paper. However, because of limited space, we’ll revise the supplementary and add illustrative examples of basic page elements in a subsection of Section D (currently page 22).

---

> > ### Comment · Area_Chair_qfKf · 2024-08-29
> >
> > Can Reviewer y5EV please comment on the authors' rebuttal? Thank you.

---

> > ### Comment · Reviewer_y5EV · 2024-08-30
> >
> > After reviewing the rebuttal and considering the feedback from other reviewers, I am inclined to maintain my original score.

---

### Official Review · Reviewer_VY5N · 2024-07-19
**Handwritten Arabic Dataset for Cursive Text Recognition**

**Rating:** 7
**Confidence:** 5
**Clarity:** The paper is well structure and easy …

**Review:**

- The paper introduces a new dataset aimed at advancing handwritten recognition of creative scripts, particularly Arabic.
- A CNN-BLSTM networks were trained with Connectionist Temporal Classification (CTC) loss for text line recognition, though other SOTA  OCR methods in other script ( e.g Chinese, and Amharic and other Latin scripts)  are need to be evaluated.
- The dataset is extensive and well-curated, providing valuable data that contribute significantly to the field of handwritten text recognition, especially for Arabic scripts.
- Overall, the paper is a significant and well-written contribution, offering a valuable dataset and novel methods.

**Strengths:**

- The paper proposes a novel Historical Arabic handwritten recognition dataset that can be used for various tasks such as layout detection, recognition, and line segmentation.
- The dataset is a rich collection of a wide range of images with different characteristics. It includes diverse writing styles on various paper backgrounds, ranging from clear handwriting to instances of torn paper or ink bleeds, capturing the complexities and variations observed in real-world Arabic handwritten manuscripts.
- Although they evaluate with a single model, baseline results reported shows the dataset's usefulness and its significance as a reference for future research.
- The paper provides an easy-to-use graphical interface for annotating the dataset, facilitating the annotation process.

**Additional Feedback:**

See above

**Correctness:**

As this work is a dataset paper, the authors have provided a clear description of the data collection and preparation process. However, it would be beneficial to evaluate using other  SOTA methods that have been applied to similar scripts

**Documentation:**

The authors have provided a description of the intended use and access for the dataset in their GitHub repository and  prepared a datasheet, and to provide clear and detailed information about the data creation process and maintenance plan.

**Ethics:**

As the dataset consists of various historical manuscripts, there is a potential risk of misuse of personal data

**Limitations:**

The authors have not explicitly disclosed the limitations of their work

**Opportunities For Improvement:**

- It would be beneficial to evaluate and provide the agreement between non-Arabic speaker annotators and the Arabic expert reviewer, comparing the text before and after expert review.
- Evaluating the dataset with SOTA OCR methods in other similar script e.g Amharic script and other would enhance the generalizability and comparative analysis of the dataset.
- Including writer information, if possible, would be helpful for tasks such as writer identification
- Providing results using normalized edit distance, e.g  see the competitions like https://rrc.cvc.uab.es/?ch=12&com=tasks, would offer a more standardized metric for evaluating text recognition accuracy.

**Relation To Prior Work:**

The paper clearly describe previous works and related dataset and briefly discuss its relation with the current work.

**Summary And Contributions:**

The paper presents a new dataset, Muharaf, for handwritten Arabic script, consisting of about 1,600 historical Arabic handwritten pages. Each page image is transcribed by experts in archival Arabic,  and a spatial polygonal coordinates of its text lines are provided. The documents include handwritten manuscripts such as personal letters, diaries, notes, poems, church records, and legal correspondences. Additionally, the paper offers baselines using a CNN-BLSTM based model. The paper is well-written and easy to follow

---

> ### Author Rebuttal · Authors · 2024-08-15
>
> We thank Reviewer VY5N for appreciating our paper and providing valuable feedback. We will revise our manuscript based on the provided comments and suggestions. Below is our response to the points raised in the review.
>
> # Evaluate agreement between non-Arabic vs. Arabic expert’s annotations
> We would like to clarify that no non-Arabic speaker was involved in the transcription step. The transcription step involved typing the text corresponding to a line image. If the transcriptions were predicted by the start, follow, read system, then the transcriptions were corrected by expert Arabic speakers. All transcriptions went through a QA cycle before being finalized.
>
> For the first 180 images, only the text lines were segmented by a non-Arabic speaker, whose first language is Urdu. As Urdu is written using the Arabic script, the annotator was familiar with various characteristics of the Arabic script and was able to mark the individual lines of text. The transcriptions/text was entered by a history professor for these lines.
>
> For 1400+ images both annotations and transcriptions were performed by expert Arabic speakers. Section 3 of our paper (on Page 5) details the entire annotation process.
>
> # Evaluation on similar scripts
> To evaluate our system, it would be more relevant to run experiments on an “Arabic-like” script. The Amharic script (written left to right) differs from the Arabic script (written right to left) in terms of its character set and various other attributes. Our trained model would not work in its current form on this script. Hence, we ran a cross dataset evaluation on the KHATT dataset. KHATT dataset is composed of handwritten samples of modern day writers, whereas Muharaf has historical handwritten samples.  Please see Table 3 of the attached PDF.
>
> # Writer information
> We mentioned in Section 6 (page 9) that the identification of writers is an ongoing process. All images of Muharaf dataset are packaged with a corresponding JSON and XML file. If the writer of a document is known, then the writer’s name appears in the corresponding JSON file under the ‘writer’ key. Similarly, the writer’s name appears as meta-information in the XML file. Please see Section A of the supplementary material, where the JSON format is described. We’ll also update Section A’s description to describe the ‘writer’ key.
>
> Please note, it is not possible to identify a writer in many cases, e.g., a legal correspondence or a church record, where a scribe wrote the document on behalf of an authority or another person.
>
> # Normalized edit distance
> The CER and WER used in our paper are indeed normalized edit distances. The Levenshtein distance mentioned in Section 5 on page 8 is the edit distance. We’ll clarify this to all readers by including the name “edit distance” in Section 5.
>
> # Limitations
> We discussed the limitations of our work in Section 6 on Page 9. For the revised paper, we’ll explicitly add a subsection describing the limitations of our work.
>
> # Ethics and misuse of personal data
> Please see our response in the global rebuttal.

---

> > ### Comment · Area_Chair_qfKf · 2024-08-29
> >
> > Can Reviewer VY5N please comment on the authors' rebuttal? Thank you.

---

> > ### Comment · Reviewer_VY5N · 2024-08-30
> >
> > Thank you for addressing some of the comments. However, I still have a few remaining questions:
> >
> > - We agree that CER and WER are normalized distances. The question is how they normalized? You noted  about this normalization on line 221, which is good. However, there is a concern raised on line 229 where it is mentioned that "CER/WER for these lines can jump to as high as 1.0." How do you address this issue? The ICDAR paper I referenced suggests that NED might offer a solution for this kind of problem.
> > - I would like to see the differences in transcription text before and after the expert review/QA check in terms of CER/WER or any other preferred metrics. This comparison would help in understanding the impact of the review process on the accuracy.
> >
> > - Regarding state-of-the-art OCR models, my question is about testing your dataset on other OCR models, rather than using other datasets to evaluate your model. Since your dataset is a major contribution, it should be evaluated with multiple SOTA OCR models
> >
> > - As a side note: there is no need to change "Levenshtein distance" to "edit distance" in Section 5, as the context from your citation is clear.
> >
> > I would prefer to keep the evaluation score as it is.

---

> > > ### Author Response · Authors · 2024-08-30
> > > **Response to Official Comment by Reviewer VY5N**
> > >
> > > Dear reviewer,
> > >
> > > We thank you for your constructive review and comments. Here are our responses:
> > >
> > > # Normalized CER
> > > We normalized the edit distance using the ground truth length, which is the standard for computing CER, e.g., please see [14, 30]. Normalizing by ground truths is also the implemented in the Hugging Face library for computing CER (Please see [CER computation at HuggingFace](https://huggingface.co/spaces/evaluate-metric/cer/blob/af39b3b7914ffb126ece472884f75033cfc4727b/cer.py) ). For line-level results we average our results over the CER of all lines.
> > >
> > > The NED in the link to ICDAR you sent has normalization based on maximum(ground_truth_length, predicted_string_length).
> > >
> > > Using this NED method ensures that CER does not go above 1.0. If we normalize using  NED, our CER/WER will be slightly lower than what we are reporting in the paper.
> > >
> > > # Transcription differences before and after QA
> > > This is a great idea. We can add this in a subsection of the supplementary material to give users an idea of the effectiveness of the QA process.
> > >
> > > # State-of-the-art OCR models
> > > Thank you for the suggestion. We plan to evaluate the dataset with multiple SOTA in our future work.

---

### Official Review · Reviewer_8PxJ · 2024-07-24
**Review of Arabic HTR resource**

**Rating:** 7
**Confidence:** 3
**Correctness:** Yes
**Clarity:** Yes

**Review:**

This is an interesting and valuable resource, but literature and selected models could have been discussed much more.

### Pros
* HTR resources are relatively scarce due to their difficulty to develop, making this dataset another important resource.
* Slightly larger than other Arabic datasets, making this one of the largest ones available.

### Cons
* Results are not analysed in terms of existing literature, they are simply stated, making it difficult to understand the significance of the presented resources. Indeed, having a point of reference is important, but other studies exist in Arabic HTR and their results could have been compared against to yield an insightful analysis. Also, I wonder what the CER/WER would have been on these test data if you had used other datasets for training (will reveal the importance of this dataset compared to existing ones).
* Model selection was unclear. What is the state of the art in HTR and how is the architecture selected for benchmarking related to it? Also, why did you opt only for full-page HTR and didn't try standard line-based solutions?

**Strengths:**

The resource is the strength of this submission, given the difficulty of producing transcriptions and that this is a relatively large resource.

**Additional Feedback:**

* Table 1 should break into more columns (as in Table 2) and the caption should be improved to make this table self-explainable.
* L63: fewer > lower/smaller/etc?
* Tables: please expand the captions with more information.

**Documentation:**

Yes.

**Limitations:**

* The authors trust GPT for summaries and keywords, but they could undertake a study to assess the outcome.
* The data are based on publicly available sources; hence, it is possible that a pre-trained model used for HTR has "seen" the test data. The authors should either elaborate on why this isn't true or explain how this could be avoided.

**Opportunities For Improvement:**

* Comparison with related work in terms of contributions and experimental findings.
* The model selected for the benchmark is not discussed and so is the fact that only a single (full-page) model was used.

**Relation To Prior Work:**

Needs to be improved and the results should also be compared to those from other studies.

**Summary And Contributions:**

This study describes the development of a new Arabic dataset of images and respective transcriptions, which was benchmarked for (full-page) HTR. Obtaining transcriptions is difficult, making this an important resource for Arabic HTR and related tasks.

---

> ### Author Rebuttal · Authors · 2024-08-15
>
> We sincerely thank Reviewer 8PxJ for their positive feedback and insightful review of our paper. We will make revisions according to the comments provided. Below is our response to the issues raised in the review.
>
> # Model selection and comparison with state of the art
> One of the goals of our work is to enable OCR of languages in a low resource setting. Keeping this goal in view, we chose Start, Follow, Read, end-to-end full page handwriting recognition system (SFR) for carrying out baseline experiments (the system is explained on Section 5 on Page 8). Each network in this system can fit on an 8GB card (mentioned on Page 2 of Section 1 of our paper), making this system ideal for a low resource environment. State-of-the-art models like transformers cannot be realistically deployed in a resource constraint environment.
>
> We can take the example of a transformer based model like TrOCR, which has 62M, 334M or 558M parameters depending upon whether it is the small, base, or large model [14]. These models have a CER of 4.22%, 3.42%, and 2.89% respectively on the IAM dataset. In contrast the CNN based SFR model has a CER of 6.4% but comparatively fewer parameters. The SOL, LF, and HW networks of the system have 9M, 5M, and 18M parameters respectively. Moreover, all three of them can be loaded and trained independently. Therefore, by choosing a more traditional CNN-based network, we are willing to trade a slight reduction in accuracy for a smaller, less resource-intensive model. Table 2 of the attached PDF summarizes these results.
>
> The main contribution of our paper is the Muharaf dataset, which we want to make available to the research community. Publishing this dataset will encourage other researchers to engage with it and test the limits of their OCR systems.
>
> # Results in terms of existing literature and training on existing datasets
> Table 2 of the attached PDF reports results from existing literature for the IAM dataset. We conducted additional cross-dataset evaluation with the KHATT dataset. The results are summarized in Table 3 of the attached PDF. We can see that when the Start, Follow, Read (SFR) system is trained with Muharaf alone or a combination of RASAM and RASM datasets, the CER is 38% and 41% respectively. However, training with all three datasets reduces the CER to 24%. While this is still higher than the transformer’s 18% CER, it shows that Muharaf plays a significant role in improving the model's performance when combined with other datasets. Moreover, the comparison with transformer architecture is not fair as the transformer was trained on the KHATT dataset and has 153M parameters compared to SFR’s HW network with only 18M parameters.
>
> # Full-page HTR vs. standard line-based solutions
> We conducted both full page HTR and line-level HTR. For page level OCR, the system detected the individual line and then predicted its text. For the line level experiments we took the ground truth annotation of each text line and ran the handwriting prediction network on it.  Please see Table 3 (on page 8) of our paper, which lists both page-level and line-level results.
>
> # GPT for summaries and keywords
> We generated summaries and keywords from OpenAI’s GPT 3.5 model for the general interest of the research community who are non-Arabic speakers. However, we don’t claim that these summaries and keywords are 100% correct. In the supplementary material, we’ll add a clarification in Sections B and C, where the various collections are discussed and the keyword generating procedure is outlined.
>
> # Pretraining the model
> We have provided the details of the pretrained model on our Github webpage. We pretrained the HW model using synthetically generated Arabic lines. The initial SOL and LF models were trained from scratch with random weights. The text of the Arabic lines was taken from the Khaleej-2004 newspaper corpus [31].  We also plan to make the synthetic images used for pretraining publicly available. In short, there is no chance that the HTR model could have "seen" the test data, neither the images nor their corresponding transcriptions. We’ll revise the supplementary material and add a subsection to explain the pretraining process.
>
> # Table 1: Break into more columns
> In the revised paper, we’ll break up Table 1 into different columns including vocabulary, total writers (if applicable/known), and comments.
>
> # L63
> We’ll change 'fewer’ to ‘less’.
>
> # Adding more captions to tables
> In our revised paper, we will add more captions to tables to provide more information to readers.

---

> > ### Comment · Area_Chair_qfKf · 2024-08-29
> >
> > Can Reviewer 8PxJ please comment on the authors' rebuttal? Thank you.

---

> ### Comment · Reviewer_8PxJ · 2024-08-30
>
> I thank the authors for their response and the respective updates. I will update my score to 8.

---

### Official Review · Reviewer_9rc2 · 2024-07-24
**Muharaf: Manuscripts of Handwritten Arabic Dataset for Cursive Text Recognition**

**Rating:** 5
**Confidence:** 3
**Clarity:** Yes, it is well written.

**Review:**

The submission presents the Muharaf dataset with thorough detail on its creation, structure, and potential applications. The authors describe the data acquisition and annotation process. While the submission is generally well-structured, it lacks some clarity in specific areas. The description of Arabic scripts and historical variations is insufficient, especially for readers unfamiliar with Arabic. Terms like "harakat" are used without explanation or examples, which can be confusing for non-Arabic speakers. Additionally, the labeling process could benefit from more detailed explanations and a clear breakdown of the annotators' qualifications and contributions. There are some discrepancies in the annotation process.

The Muharaf dataset is a good contribution, being one of the largest publicly available datasets for historical Arabic manuscripts. Its comprehensive annotation and transcription make it a valuable resource for advancing Arabic HTR. However, the originality could be better highlighted by directly comparing the Muharaf dataset to existing datasets and emphasizing its unique features in section two and elaborating beyond the table.

Pros:
1 - The Muharaf dataset includes a wide range of handwritten Arabic documents, offering good content for research.
2 - A significant portion of the dataset is publicly available.
Cons:
1- The paper lacks detailed discussions on the historical variations of Arabic calligraphy and scripts. Important distinctions, such as the use of diacritics and different writing styles, are not adequately addressed. For example, historical Arabic writing often omits dots, significantly altering letter forms.
2 - Terms like "harakat" are used without explanation or examples, making the paper less accessible to non-Arabic speakers. There is insufficient discussion on the type of Arabic used (Modern Standard Arabic vs. Dialects) and the historical context of the documents.
3-  While the survey of existing datasets is comprehensive, the paper does not clearly differentiate the Muharaf dataset from others. The relationship between their work and existing datasets is not well-articulated.
4 -  The description of the labeling process is vague and potentially misleading. It would be beneficial to include a table detailing who annotated what, their qualifications, and the number of annotated documents.
5 - The dataset benchmarking is not robust, limiting the assessment of its performance and applicability. Stronger evaluation results and comparisons with existing datasets would enhance the paper's impact.

**Strengths:**

The submission's primary strength lies in the Muharaf dataset's wide range of handwritten Arabic documents, which provides a good resource for advancing research in handwritten text recognition and related fields. Its significance is further underscored by the inclusion of diverse document types and writing styles, enhancing its applicability across various research domains. The dataset's public availability promotes transparency and accessibility.

**Additional Feedback:**

Strengthening the benchmarking with more comprehensive evaluations and comparisons to existing datasets will enhance the dataset's impact. Clarifying the historical and linguistic context and providing guidelines for responsible use will further improve the submission's overall quality and applicability.

**Correctness:**

The claims made in the submission regarding the creation and utility of the Muharaf dataset appear to be correct.

**Documentation:**

The paper should be clear and consistent in the annotation method. It is a bit vague and inconsistent.

**Ethics:**

No, there are no potential ethical concerns .

**Limitations:**

The authors have not fully addressed the limitations and potential negative societal impacts of their work. They do not provide sufficient discussion on the historical and linguistic complexities of Arabic scripts, which is crucial for a dataset dealing with diverse and historical handwritten documents. Additionally, the labeling process lacks transparency, and the benchmarking results are weak, limiting the assessment of the dataset's quality and applicability.

**Opportunities For Improvement:**

The submission's limitations include a lack of detailed discussion on the historical variations and specific characteristics of Arabic scripts, which is crucial for understanding and processing such diverse manuscripts. The absence of explanations and examples for non-Arabic terms and concepts, like "harakat," makes the work less accessible to non-Arabic speakers and potentially limits its broader relevance. The comparison with existing datasets is insufficient, as the paper does not clearly highlight how the Muharaf dataset differs from or improves upon previous collections. The labeling process description is vague, lacking clarity on the qualifications and contributions of annotators, which could impact the perceived quality of the research. Lastly, the benchmarking is weak, with limited evaluation results and comparisons, making it difficult to assess the dataset's performance and significance fully.

**Relation To Prior Work:**

The submission does not clearly discuss how this work differs from previous contributions. While the paper includes a survey of existing datasets and describes the Muharaf dataset's features, it falls short in explicitly contrasting these features with those of prior datasets.

**Summary And Contributions:**

The submission introduces the Muharaf dataset, a  collection of 1,644 historic handwritten Arabic manuscript images, each annotated and transcribed by experts in archival Arabic. The dataset aims to advance handwritten text recognition (HTR) for Arabic and other cursive scripts. The dataset encompasses various handwriting styles and document types, including personal letters, diaries, notes, poems, church records, and legal correspondences. Key features include spatial polygonal coordinates of text lines and basic page elements, making it suitable for tasks beyond HTR, such as text-line segmentation, layout analysis, and writer identification. The paper details the data acquisition pipeline, and notable features, and provides baseline results from training convolutional neural networks. The Muharaf dataset is a contribution to the field, offering the largest publicly available Arabic dataset for historical manuscripts, and includes tools for annotating and transcribing Arabic texts.

---

> ### Author Rebuttal · Authors · 2024-08-15
>
> We appreciate Reviewer 9rc2 for their detailed critique and feedback. We have carefully considered the comments and plan to incorporate the respective changes into the revised manuscript. Below are our responses.
>
> # Characteristics and historic variations of Arabic script
> The Arabic alphabet is believed to have its roots in the Nabatean alphabet from the third century [29]. The study of the origins of Arabic script and its evolution from classical Arabic to modern standard Arabic is a huge undertaking and still a subject of open scientific study and debate [28]. It is true that the early Arabic writings did not contain dots (ijam), however, the Muharaf dataset has writing samples from the early 19th century to early 21st century, a period during which the use of dots was well established. Most of the samples are in the Arabic script Ruqʿah. Ruqʿah became the most common style in the late Ottoman period and early post-Ottoman period in the area we know today as the Middle East (which includes Egypt, but not North Africa).
>
> We will revise our manuscript to detail the characteristics of the Arabic script relevant to the Muharaf dataset and to address the challenges associated with its recognition. We list a few points below:
>
> 1. The shape of each character of the Arabic script depends upon its contextual position within a word. Many characters have four different shapes depending upon whether they are in their isolated, initial (at beginning of a word), medial (in the middle of a word) or final forms (at the end of a word). This poses a challenge for the OCR system that has to recognize all the different forms of the same character. Please see Table 1 of the attached PDF for an example of 4 Arabic characters.
>
> 1. Additional symbols in Arabic script include ijam, which are dots present above or below a character. Two characters may have the same basic shape but different number of dots to tell them apart. For example, in Table 1 of the attached PDF, ijam or dots distinguish the bāʼ (one dot below) from tāʼ (two dots above). An HTR system can misclassify a character because of the ijam.
>
> 1. Arabic language also has diacritics, which include ijam and tashkil. Tashkils are also called harakat. Harakat are short vowel marks in Arabic and are used to indicate the pronunciation of words. They are optional symbols and may not be present in the script. For example, the phrase without diacritics: مجموعة بيانات محرف رائعة turns into مَجْمُوعَةُ بَيَانَاتِ مُحَرَّفٍ رَائِعَةٌ with diacritcs. The diacritics in the later phrase are the accent marks above or below the characters. In the context of an HTR system, the diacritics can get mislabeled for ijam and vice versa. They also increase the size of the character set that the system has to deal with.
>
> # Contrasting features of Muharaf and other publicly available Arabic datasets
> Please see our response in the global rebuttal section.
>
> # Annotators involved in the labeling process and their qualifications
> Most of the annotations and transcriptions (1400+ images) were carried out by Arabic speakers who manage digital archives of Arabic manuscripts and facilitate the information access from a large number of Arabic documents. Their exact designations are:
> - Head of the reference and external relations at Phoenix Center for Lebanese-Historical Archives (PCLS), USEK.
> - Head of storage division at digital development center, USEK.
>
> The transcriptions of both individuals were checked by a full professor of history who is also a historian and the director of USEK Library, Phoenix Center for Lebanese Studies.
>
> Some transcriptions (around 180 images) were carried out by
> - Assistant professor teaching history at the center for Lebanese studies
>
> These transcriptions were checked by a full professor of history at KCLDS, NC State.
> We’ll add this information in the main paper as well as the datasheet of the supplementary material (page 25).
>
> Please note that each image of the packaged Muharaf dataset is accompanied by a JSON and XML file. The JSON and XML files contain the names of the annotator and the transcriber of that image. In our revised supplementary material, we'll create a summary table in the datasheet to provide the exact numbers as suggested by the reviewer.
>
> # Evaluation results and comparisons
> We have included additional evaluation results and comparisons for the IAM and KHATT datasets in the attached PDF. The performance of a transformer based system TrOCR is compared with the Start, Follow, Read, end-to-end full page handwriting recognition system (SFR) for IAM in Table 2. While the performance of TrOCR is better than SFR, it has a higher computational complexity and requires more resources. Our choice of SFR is based on the goal to develop a system for a low resource environment.
>
> Table 3 includes the results of cross-dataset validation on the KHATT dataset. When SFR is trained with Muharaf or RASAM+RASM datasets, the accuracy of both trained models is not very good. However, when all three datasets are combined, the system’s CER improved by ~16%, showing the significance of Muharaf as a valuable addition to the current publicly available Arabic datasets.
>
> # Limitations and potential negative societal impacts
> We discussed the limitations of our work in Section 6 on Page 9. For the revised paper, we’ll explicitly add a subsection describing the limitations of our work. There are no negative societal impacts of our work. We marked item 1c of the paper’s checklist on page 12 as N/A to clarify that this work does not have any negative societal impacts.

---

> > ### Comment · Area_Chair_qfKf · 2024-08-29
> >
> > Can Reviewer 9rc2 please comment on the authors' rebuttal? Thank you.

---

### Author Rebuttal · Authors · 2024-08-15

We thank all the reviewers for their thorough critiques and valuable feedback on our paper.

# Ethics review: privacy, legal or ethical issues
**Response to Ubyq, VY5N, and RbDt’s Q3:** The images of Muharaf-public were already publicly accessible before our proposed redistribution effort; so anonymization is unnecessary and there is no concern regarding the leakage of personal information. The images were available in the archives at USEK or NC State. Readers could access the images through the centers’ websites or as hard copies through the libraries.

**Response to RbDt’s Q1:** The authors are the members of the two archives and have the rights to redistributed images as a part of Muharaf-public.

**Response to RbDt’s Q2 and 9rc2:** The limited licensing is due to the proprietary nature of the images, and hence are authorized to use only with permission from the owners. The usage restrictions include that the dataset shall be used solely for research purposes and not be redistributed without permission.

**Response to RbDt’s Q4:** We will require researchers to agree to the statement of ethical use of data as a requirement of downloading the data. In addition, the research process described in this manuscript sticks to the ethical standard.

We plan to incorporate all above points into the revised manuscript.


# Contrasting features of Muharaf and other publicly available Arabic datasets
We’ll revise our paper to include a comparison of Muharaf with other datasets. The existing Arabic datasets are a valuable resource for the research community. Muharaf supplements them with its own unique features and characteristics, discussed below:

1. As shown in Table 2 on page 5 of our paper, Muharaf has the largest number of line images as compared to IAM, RASAM, RASM, and KHATT.
2. KHATT and MADCAT datasets are category 1 (scribed) datasets, where writers were given text to write under controlled experimental conditions. Muharaf is category 2 (original) dataset, where original historic manuscripts were scanned and transcribed.
3. RASAM and RASM are two Arabic datasets of scanned original historic manuscripts from category 2. The main difference between these two datasets and Muharaf lies in the Arabic script with which they were written (see points 4 and 5 below). Another difference is that RASAM and RASM handwritten pages belong to books, with calligraphic handwriting that is very neat and uniform across pages, written in straight horizontal lines. They were written by scholars of their respective fields. Muharaf includes informal/personal styles of writings, which were very common in the 19th and 20th centuries. The samples vary from very neat to barely legible writing. The handwriting samples of the same individual can be different over different documents or letters. The text lines can be slanted upward or downward instead of horizontally straight lines.
4. RASAM has three types of manuscripts from the 10th century. They are scanned pages of books, which were written in the “Meghrebi script” also known as the “Round script” [28]. As the name suggests, this script has very rounded shaped letters. In contrast, Muharaf’s documents are mostly Ruqʿah script, which is used for everyday or casual writing. It is composed of straight, short lines, and simple curves.
5. RASM has 4 different types of manuscripts of scientific writings from the 8thcentury to the 19th century. Muharaf has images from 50 different collections, each collection having one or more writers. Like RASAM, RASM’s handwriting styles are calligraphic, very neat and uniform across all pages as opposed to Muharaf, where the writer may not have very careful or readable handwriting.

Please note, that the OCR of text from different Arabic datasets like RASAM, RASM has its own challenges and by no means an easy feat. We intend to supplement the existing Arabic datasets with a variety of handwritten images with the goal of digitizing handwritten documents from the late 19th century to the mid-20th century.

---

### Decision · Program_Chairs · 2024-09-26

**Decision:**

Accept (Poster)

**Comment:**

This paper received 1 negative review and 3 positive reviews, with the average rating of 6.25. The rebuttal comments did not convince all reviewers, who did not raise their initial ratings. The low average rating places this paper below the acceptance threshold. However, one reviewer promised to raise their score to 8, but forgot to do so. Therefore, the paper should be accepted.